# Three-Year Real-World Outcomes of Interwoven Nitinol Supera Stent Implantation in Long and Complex Femoropopliteal Lesions

**DOI:** 10.3390/jcm12144869

**Published:** 2023-07-24

**Authors:** Szymon Salamaga, Hubert Stępak, Mikołaj Żołyński, Jagoda Kaczmarek, Maciej Błaszyk, Michał-Goran Stanišić, Zbigniew Krasiński

**Affiliations:** 1Department of Vascular and Endovascular Surgery, Angiology and Phlebology, Poznan University of Medical Sciences, Długa Street, 61-848 Poznan, Polandmichal.stanisic@skpp.edu.pl (M.-G.S.); zbigniew.krasinski@gmail.com (Z.K.); 2Department of Cardiology-Intensive Therapy, Poznan University of Medical Sciences, Ul. Przybyszewskiego 49, 60-355 Poznan, Poland; 3Faculty of Medicine, Poznan University of Medical Sciences, 1/2 Długa Street, 61-848 Poznan, Poland; 4Department of Radiology, Poznan University of Medical Sciences, 1/2 Długa Street, 61-848 Poznan, Poland; maciej.blaszyk.jr@gmail.com

**Keywords:** peripheral artery disease, chronic limb ischemia, atherosclerosis, Supera stent, femoropopliteal, complex lesions, long lesions

## Abstract

Background: Peripheral artery disease (PAD) remains a major issue in modern societies and affects more than 200 million people around the world. Endovascular methods have been evaluated to be safe and effective in limb salvage. The Supera is able to withstand increased compression, biomechanical stress and to have higher radial force. The objective of this study is to evaluate performance, durability and 3-year patency of Supera stent implantation in severe femoropopliteal disease. Methods: A retrospective real-world analysis was performed with consideration of 77 patients that had a Supera stent implanted with femoropopliteal atherosclerotic disease at a single center. Among the 77 individuals, 92 Supera stents were implanted. Analysis of patients’ demographics, lesions characteristics, reintervention rates and patency rates was performed. Results: The median follow-up was 33 months and ranged from 0 to 84 months. Chronic limb-threatening ischemia was observed among 43 patients. Mean lesion length was 152.8 ± 94.6 mm. Chronic total occlusions were observed in a majority of lesions. Overall, primary patency rates at 6, 12, 24 and 36 months were 85.0%, 73.6%, 59.2% and 53.2%, respectively. Conclusions: The Supera stent is effective in the management of long and complex lesions. The results of patency rates were evaluated to be worse among lesions extending to the popliteal artery.

## 1. Introduction

Peripheral artery disease (PAD) remains a major issue in modern societies and affects more than 200 million people around the world [1]. Atherosclerotic disease in a femoropopliteal location, causing significant stenosis or occlusion of the superficial femoral artery (SFA) and/or the popliteal artery (PA), can lead to life-limiting intermittent claudication (IC) or pain at rest and non-healing ulcers. Rest pain and non-healing ulcers are included in chronic limb-threatening ischemia (CLTI). CLTI is associated with increased rates of major adverse cardiovascular events (MACE) and major adverse limb events (MALE) [2,3]. To prevent those aggravating complications, proper treatment is required. Nonetheless, the latter bears a considerable mortality in the short and medium term, given the high incidence of risk factors for atherosclerosis and coronary artery disease in patients with CLTI [4].

Endovascular methods were evaluated to be safe and effective in limb salvage. Current guidelines indicate the increasing role of endovascular methods in therapy of femoropopliteal lesions [5,6]. Previous guidelines favored percutaneous transluminal angioplasty (PTA) as the primary preferred method of endovascular treatment of femoropopliteal lesions with bail-out stenting placement when the optimal result was not achieved [7]. However, more recent studies suggest that initial stenting, especially in long and complex lesions in the SFA, leads to superior results with higher patency rates [8]. Nevertheless, endovascular treatment in femoropopliteal region remains challenging, due to forces that interact in this location. Femoropopliteal artery is exposed to extensive compressive forces caused by knee and hip joints mobility during locomotion [9,10,11]. Different types of stents were designed for implantation in SFA and PA, however not all achieved satisfactory results. Mechanical stress causes the risk of stent fatigue and fracture [12]. Complexity and length of the target lesions is associated with lower patency rates and severe lesions remain a challenge in treatment [13].

The Supera stent (Abbott Vascular, Santa Clara, CA, USA) is a braided, interwoven nitinol stent with helical construction. The Supera self-expanding stent mimics anatomical natural movements in SFA and PA [14]. Thanks to its unique design, the Supera stent is enabled to withstand increased compression, biomechanical stress and to have higher radial force [15,16].

The objective of this retrospective, real-life study is to evaluate performance, durability and 3 years patency rates of Supera stent implantation in severe femoropopliteal disease. The material covers real-world results from one center.

## 2. Materials and Methods

This is a physician-initiated, single-center, observational cohort study with a retrospective analysis of the collected data. A retrospective, single-center analysis was performed with consideration of patients that had a Supera stent implanted in femoropopliteal atherosclerotic disease. All patients with chronic limb ischemia that had been treated with Supera stent implantation in the severe lesions in femoropopliteal localization between 2012 and 2019 were included. The inclusion criteria to the analyzed cohort were chronic limb ischemia, Rutherford class 2–6, Supera stent implantation in femoropopliteal region. Two patients with Rutherford class 2 were included, those were active and young (48 and 54 years old) patients with moderate claudication that limited their functioning. The only exclusion criteria was Supera stent implantation in femoropopliteal bypass. The selection of devices, chosen strategy and duration of antiplatelet therapy were at the discretion of performing specialist. The study was conducted in accordance with the Helsinki Declaration. All patients signed a written consent form before receiving endovascular treatment. The patients’ privacy was not violated by data collection. The study was accepted by the institutional bioethics committee.

### 2.1. Study Population

Extraction of all Supera stents implantation procedures was performed from the database. All clinical and demographic information was extracted from electronic medical system. Clinical manifestation of PAD was ranked in all patients according to Rutherford classification. Angiograms were reviewed to collect data on lesion morphology, additional procedures, reference diameter of the artery and procedural complications. Two experienced vascular surgeons independently evaluated angiographic images.

### 2.2. Procedure Details

Patients were qualified for endovascular procedure based on clinical evaluation and computed tomography angiography (CTA). All procedures were performed by experienced vascular surgeons and according to our institutional standards. All cases were performed under local anesthesia while conscious. The access was achieved using 6F sheath via contralateral or ipsilateral access. Anticoagulation using 50 IU/kg of heparin was applied after access obtainment. Initial angiography with standard contrast administration was conducted to define vascular anatomy and lesion characteristics. Intraluminal crossing was performed in every possible case. If subintimal passage was performed, the distal reentry was tested with contrast injection. A reentry device was not used in any case. After the lesion was crossed using guidewire and before stent deployment, inflation of a balloon was performed in all patients to carefully prepare the vessel for stent implantation. The size of the balloon was based on measurement of the artery diameter during angiography. The indication for Supera stent implantation was established individually by the treating vascular specialist. Reasons for stent implantation varied in different cases and included subintimal crossing, dissection, recoil, residual stenosis >30% and primary stenting that was performed in extremely calcified and complex lesions when there was a high possibility of complications. Stent diameter was selected in order to match the nominal diameter of the vessel in 1:1 ratio. Lesions were covered together with at least 5 mm of lesion-free zone. If more than one stent was needed, overlapping for a minimum of 10 mm was applied. Closure of the access puncture was achieved with either manual compression or closure device (ANGIO-SEAL^®^ VIP Vascular Closure Device, Terumo or The StarClose SE™ Vascular Closure System (VCS), Abbott). After the procedure all patients were prescribed dual antiplatelet therapy (clopidogrel + aspirin) for at least 30 days.

Non-CLTI patients in the first line were qualified for the endovascular procedure. In CLTI, the preferred option of treatment of complex femoropopliteal lesions was vein infrainguinal bypass surgery. During 2012–2019, about 70 femoropopliteal bypasses were performed every year. Nevertheless, some patients either were not suitable for open procedure, did not accept the risk of the open surgery or did not have a suitable vein. If stent implantation was obligatory after primary angioplasty and lesions extended to P2 or P3, then the Supera stent was preferred. Lesions that did not involve the P2 or P3 segment were treated with Supera stent if high mechanical stress on the stent was expected. For example, complex, long lesions with a high degree of calcification were expected to produce higher compression force.

The Supera peripheral stent is a self-expanding stent that is made of six pairs of interwoven wires that are arranged in a helical pattern. Such design enables the stent to be both flexible and resistant to fracture. Supera is provided on a 6F or 7F delivery system and the shaft lengths vary from 80 cm to 120 cm. The system can be delivered over 0.014 or 0.018 guide wires. Available stent diameters are from 4.5 mm to 7.5 mm and length from 20 mm to 200 mm. Oversizing of the stent is not required and Supera should match the reference diameter of the vessel with 1:1 ratio. Predilation of the artery should be performed before the stent implantation. Implantation of the Supera stent must be performed with special concern not to elongate the stent. What was proven in previous research is that elongation of the Supera stent leads to worse results with lower patency rates and higher rates of restenosis [15,17].

### 2.3. Definitions

Intermittent claudication was defined by pain in the calf region when walking, not occurring with sitting or standing, which does not disappear during the walk and causes the subject to stop or slow down, and is relieved within 10 min of walking cessation. CLTI was defined as rest pain, ulcer or gangrene associated with proven arterial occlusive disease. Patients were assigned to the renal insufficiency group if GFR < 60. Length of the lesions were classified as short at ≤6 cm, intermediate at 7 to 10 cm, or long at >10 cm. Calcification degree was evaluated among treated lesions and categorized as either none, mild, moderate or severe. None was defined as absence of calcification. Lesions evaluated as mild had <25% circumference. Moderate was defined as calcification of 25–50% of circumference. Severe was defined as >50% of circumference [18]. Hematoma was defined as collection of blood around the access site. The definition of pseudoaneurysm was a false aneurysm involving the access site. Chronic total occlusion was defined as complete occlusion of the arterial segments for ≥3 months duration. Analysis of patency rates was performed and they were aimed as a primary endpoints. We established definition of primary patency rate as a lack of clinical deterioration or significant restenosis evaluated in ultrasound follow-up. In ultrasound follow-up, the definition of significant restenosis was peak systolic velocity > 250 cm/s or peak systolic ratio > 2, or a monophasic waveform in the artery distal to the stent. The definition of assisted primary patency was patency rate of intervention that had to be retreated but did not fall to the level of thrombosis. Secondary patency was classified as patency rate of procedure that failed to the level of thrombosis and was retreated successfully. In the analysis of the procedures technical success was defined as achievement of reestablishment of vessel patency and residual stenosis less than 30%. Procedural success was defined as technical success and no complications during procedure. Major amputation was defined as an amputation of the lower limb on the level of the ankle or above.

### 2.4. Follow-Up

Peri-procedural complications during 24 h after implantation were analyzed. Follow-up visits at the outpatient clinic were planned for 1 month, 6 months, 12 months and then every 12 months after the procedure. On reevaluation visits, patient physical examination and duplex ultrasonography was performed. Primary outcomes conducted in this study were patency rates, including analysis of primary patency, assisted primary patency and secondary patency rates. Additionally, patency rates were calculated in groups of stents implanted to SFA, to PA and to both.

### 2.5. Statistical Analysis

Continuous characteristics are presented as mean ± standard deviation or as median (range). Categorical variables are given as count (percentage). Outcomes such as primary patency rate, primary assisted patency rate and secondary patency rate were calculated using Kaplan–Meier survival analyses with 36 months period. Standard Errors (SE) and 95% Confidence Intervals (95% CI) were calculated. Differences in Kaplan–Meier curves were evaluated between SFA, PA and SFA+PA groups. Differences between groups in patency rates were evaluated with the log-rank test. Statistical significance was conducted if the two sided *p*-value < 0.05. Statistical analysis was performed using Statsoft Statistica 13.3 software (TIBCO Software Inc., Palo Alto, CA, USA, 2017).

## 3. Results

Seventy-seven patients were included in this retrospective single-center study. Among 77 consecutive individuals, 92 Supera stents were implanted. Demographics of analyzed patients were summarized in Table 1. The most common comorbidities were arterial hypertension (81.8%) and diabetes mellitus (54.1%). Additionally, 72.7% of patients reported smoking. Lesion characteristics and used devices details are presented in Table 2.

Stratification according to Rutherford classification was performed. Most individuals presented with class 3 (40%) manifestation of LEAD. In addition, ten patients presented with rest pain (Rutherford class 4), 34 (44%) patients had tissue loss (Rutherford class 5 and 6). Forty-four patients were classified as CLTI, which accounted for 57% of the cohort.

Among 77 procedures, 19 interventions were performed due to in-stent restenosis (ISR), which accounted for 25% of the lesions. Of the lesions, 20% required more than one stent to be implanted; the implantation of three stents was not required in any case. Mean lesion length was 152.8 ± 94.6 mm. Chronic total occlusions were observed in a majority of lesions, accounting for 66%. Severe and complex lesions accounted for the majority of interventions. According to the Trans-Atlantic InterSociety Consensus Document (TASC II) classification, most of the lesions were classified as TASC II C or D (65%). Among all 92 implanted stents, median diameter of the stent was 5.5 mm and median length of the stent was 150 mm. The most common stents used in the procedures were 5.5 mm (33%) and 4.5 mm (22%) in diameter.

Technical success was achieved in 97.4% of the procedures. In one case, despite post-dilatation with the balloon, the residual stenosis remained >30%, and in another case, the stent was damaged and another Supera stent implantation was required. Procedural success was achieved in 90%. In terms of complications, no MACE or MALE were observed during 30-day follow-up. However, minor complications were noted: four dissections of the artery, two hematomas in access area, one popliteal artery bleeding and one distal embolism.

The median follow-up was 33 months and ranged from 0 to 84 months. For more than 12 months, 67.5% of the individuals remained in the follow-up; 53.2% of patients remained in follow-up for more than 24 months and 49.4% of individuals remained in follow-up analysis more than 36 months. Overall patency rates are summarized in Figure 1. Overall primary patency rates at 6, 12, 24 and 36 months were, respectively, 85.0%, 73.6%, 59.2% and 53.2%. Overall assisted primary patency rates at 6, 12, 24 and 36 months were, respectively, 84.7%, 78.0%, 68.7% and 66.4%. Secondary patency rates at 6, 12, 24 and 36 months were, respectively, 98.4%, 89.6%, 83.4% and 81.2%. During registered follow-up, no stent fractures were observed. Additional analysis of the subgroups was performed; patients were divided according to localization of the lesions. Primary patency rates divided according to localization of the lesions are depicted in Figure 2. Primary patency at 36 months was 74.3%, 45.3% and 43.6% in lesions in superficial femoral artery, popliteal artery and both, respectively. Assisted primary patency rates divided according to localization of the lesions are summarized in Figure 3. Assisted primary patency at 36 months was 90.2%, 64.1% and 51.9% in lesions in superficial femoral artery, popliteal artery and both, respectively. Secondary patency rates divided according to localization of the lesions are depicted in Figure 4. Secondary patency at 36 months was 94.7%, 78.6% and 76.0% in lesions in superficial femoral artery, popliteal artery and both, respectively. During 36-month follow-up, five major amputations were performed among the analyzed cohort, all of them in patients with CLTI. Moreover, three minor amputations were performed during follow-up. Limb salvage rate at 36 months was 84.7%.

Log-rank analysis revealed statistically significant differences in patency rates between groups divided according to localization of the lesions. Primary patency rate was significantly higher in SFA group compared to the SFA + PA group. In log-rank analysis, the *p*-value was 0.02 and primary patency rates at 36 months were 74.3% and 43.6%, respectively. Primary assisted patency rate was evaluated to be significantly higher in the SFA group compared to the SFA + PA group. Log-rank analysis revealed a *p*-value of 0.01 with primary assisted patency rates at 3 years 90.2% in SFA and 51.9% in SFA + PA. Furthermore, a significant difference was observed in secondary patency rates between SFA and SFA + PA groups with *p*-value = 0.02 in the log-rank test. Reported secondary patency rates at 36 months were, respectively, 98.4% and 83.4%. There were no significant difference in patency rates between the PA group and the SFA + PA group. Primary patency rate in the PA group was significantly lower in comparison with the SFA group (*p* = 0.04 in log-rank test). The 36-month primary patency rates in the PA and SFA groups were, respectively, 74.3% and 45.3%. No differences were found in assisted primary patency (*p* = 0.07) and secondary patency (*p* = 0.06) rates between SFA and PA groups in the log-rank test.

Reinterventions are summarized in Table 3. Stent implantation due to ISR and directed thrombolysis were the most common primary reinterventions after Supera stent implantation. Stent implantation was performed in eight patients and directed thrombolysis was applied in eight individuals. Femoropopliteal bypass as a reintervention procedure after Supera stent implantation was performed in six patients during the analyzed follow-up period.

## 4. Discussion

Preservation of patency after recanalization of complex femoropopliteal lesions remains one of the biggest challenges of endovascular treatment [19]. European guidelines have established recommendations that an endovascular first strategy should be applied in femoropopliteal lesions with length < 25 cm and in femoropopliteal lesions superior to 25 cm in patients unfit for the surgery [6]. Nevertheless, the best endovascular treatment in long and complex femoropopliteal lesions is not yet well established. DURABILITY-200 and STELLA studies demonstrated primary patency rates at 12 months of 64.8% and 66.0%, respectively [20,21]. Some authors have presented results of use of directional atherectomy devices in long femoropopliteal lesions [22,23]. Artzner et al. presented results of a rotational thrombectomy device (Rotarex system, BD) in management of chronic limb ischemia. The 12-month primary patency achieved in this study was 70.5% [24]. In comparison to these data, Supera stent performance in our analysis was good, with a 12-month primary patency rate of 73.6%. Nevertheless, the comparison between these studies is difficult, as there are major methodological differences. A number of the mentioned studies present prospectively collected data [20,21,25], and others are retrospective studies [24,26]. For assessment of the gold standard in treatment of complex femoropopliteal lesions, comparative studies are required. 

The stent fracture remains a limitation in endovascular treatment of femoropopliteal lesions and is associated with worse long-term patency rates [12,27,28]. Prevalence of stent fracture is higher among long-stented segments [29,30]. In the analyzed group of individuals, no stent fractures were observed during the 3-year follow-up. Studies from the literature that included only patients with popliteal artery lesions also did not present a stent rupture despite the tendency in this localization [31,32,33,34]. In the literature, there were only few cases of Supera stent fracture [35,36].

The retrospective analysis revealed that Supera stents have good patency rates in severe and long lesions in the femoropopliteal region. There is not much evidence regarding long and severe lesions. Most of the studies conduct lesions that are shorter than 150 mm [14,17,37,38,39,40,41]. Only a few, including our study, present results of longer lesions. In the literature, authors that evaluated long, severe lesions reported primary patency rates at 12 months of 72.6–94.1% [16,42,43,44]. Additionally, most of the analyzed lesions were CTOs (66.2%); only Palena et al., Nasr et al. and Miao Yang et al. reported a higher prevalence of CTO [42,44,45]. This fact also distinguishes our study among the existing literature. Moreover, data regarding Supera stent performance in long and complex lesions with a follow-up period of more than 24 months is scarce, as most of the studies present 1-year follow up. According to TASC classification, most of the lesions in this population were classified as C or D, which indicates severity and complexity of the lesions that occurred among individuals included in this study. Severe lesions are proven to have worse outcomes [8,46,47,48]. What is more, morphology of the lesion remains the main risk factor in long-term outcomes and short stenoses have better results compared to long total occlusions [27,49]. In comparison, Nasr et al. and Van Meirvenne et al. reported primary patency at 24 months of 77.9% and 60.8%, respectively; both cohorts consisted in the majority of patients with TASC C/D lesions treated with the Supera stent [42,43]. We did not exclude patients that presented with restenosis lesions, and restenosis accounted for 24.7% of the analyzed lesions. This fact distinguishes our paper among existing literature involving patients with long lesions treated with Supera stent implantation [16,37]. 

In our study, results were good in lesions that did not involve popliteal artery and patency rates were significantly worse when lesions extended from SFA to PA. PA artery is associated with greater stress and higher flexion–extension, compression and rotational forces. No stent fractures were observed; nevertheless, greater mechanical stress supposedly leads to higher restenosis and lower patency rates. This fact was confirmed in the results achieved retrospectively in the study published by Brescia et al. [50]. Additionally, it was reported in the BEST-CLI trial and recent registries (RECCORD registry) that the rate of stenting in the popliteal segment is high (40–50%) [51,52]. Since the popliteal artery is considered as a “no stenting” zone due to the abovementioned anatomical reasons, this may have accounted for the inferior performance of the endovascular therapy in the BEST-CLI trial [52]. Worse long-term results in stenting of the popliteal artery were confirmed in this study even though interwoven stents were used and no stent fractures were observed. It is crucial to reduce the number of complications that lead to bail-out stenting in the popliteal artery, such as flow-limiting dissection or recoil. The usage of atherectomy devices may reduce barotrauma and the dissection process, which lead to higher rates of bail-out stenting [22].

Previous large cohort studies that evaluated Supera stent performance involved patients in great abundance with intermittent claudication [37,39,40,41]. In contrast, our study consists mainly of patients that presented with CLTI (57%). Predominance of CLTI in the analyzed cohort indicates worse results concerning limb salvage and survival rate [53].

The results of our study are promising; however, a choice between Supera stent implantation and different endovascular procedures as a first line treatment is still not yet established in patients with severe lesions. Nevertheless, more prospective, randomized controlled trials with homogenous groups are required.

### Limitations

The main limitations of the study are its retrospective design, lack of randomization and comparison with different method of treatment, single-center cohort and lack of standardization of procedural methods.

## 5. Conclusions

The outcomes of this study are encouraging for Supera stent utility in the management of long and complex lesions, but further prospective and randomized controlled trials are needed in order to determine the effective, routine use of this endovascular technology. Especially the comparison of Supera stent endovascular treatment with open surgical treatment and with other endovascular methods (atherectomy, lithoplasty, other novel stents implantation) in long, complex lesions is required. The results of patency rates were evaluated to be worse among patients that had lesions extending to the popliteal artery.

## Figures and Tables

**Figure 1 jcm-12-04869-f001:**
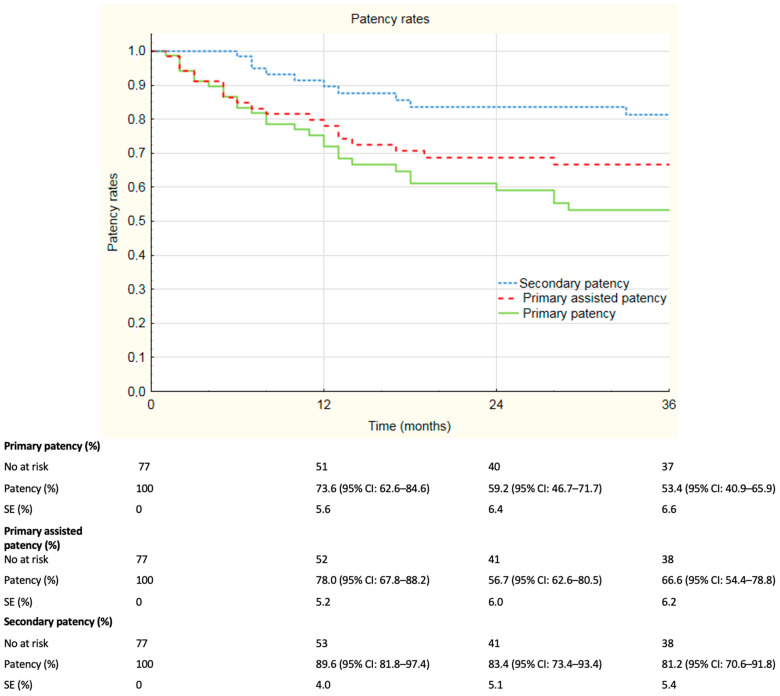
Kaplan–Meier estimates presenting overall primary patency rate, primary assisted patency rate and secondary patency rate of Supera stent implantation on duplex 36 months follow-up. SE—standard error, CI—confidence interval.

**Figure 2 jcm-12-04869-f002:**
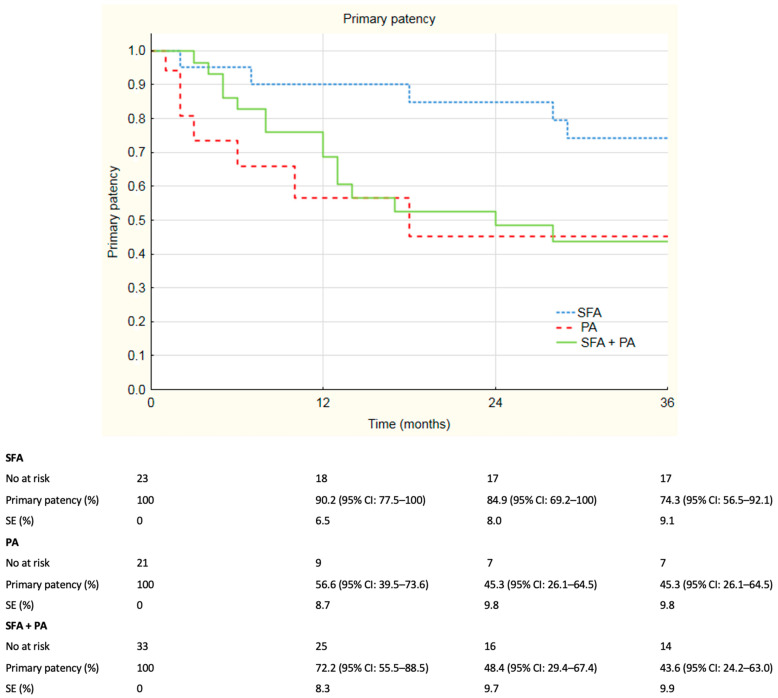
Kaplan–Meier estimates presenting primary patency rates of Supera stent implantation by localization on duplex 36 months follow-up. SFA—superficial femoral artery, PA—popliteal artery, SE—standard error, CI—confidence interval.

**Figure 3 jcm-12-04869-f003:**
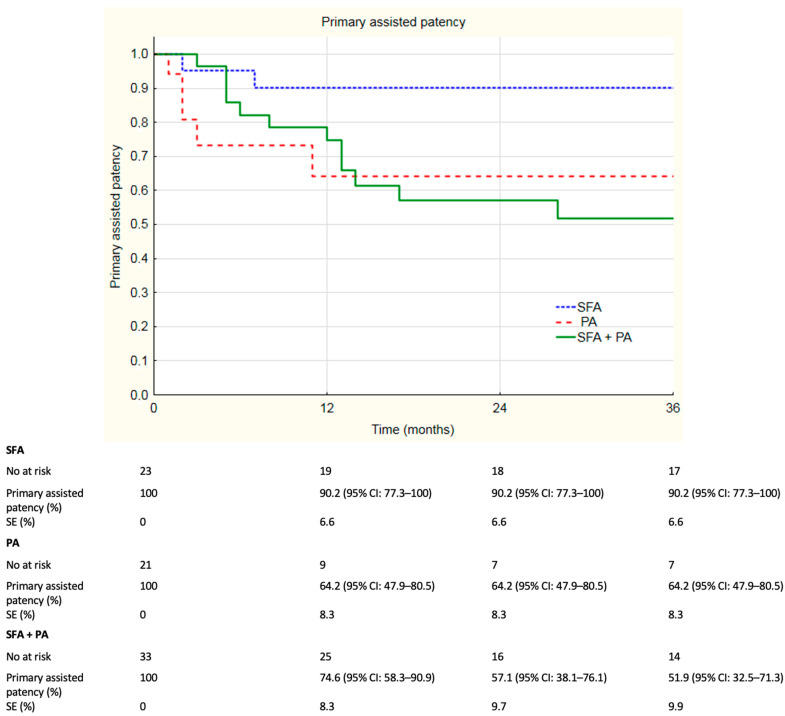
Kaplan–Meier estimates presenting assisted primary patency rates of Supera stent implantation by localization on duplex 36 months follow-up. SFA—superficial femoral artery, PA—popliteal artery, SE—standard error, CI—confidence interval.

**Figure 4 jcm-12-04869-f004:**
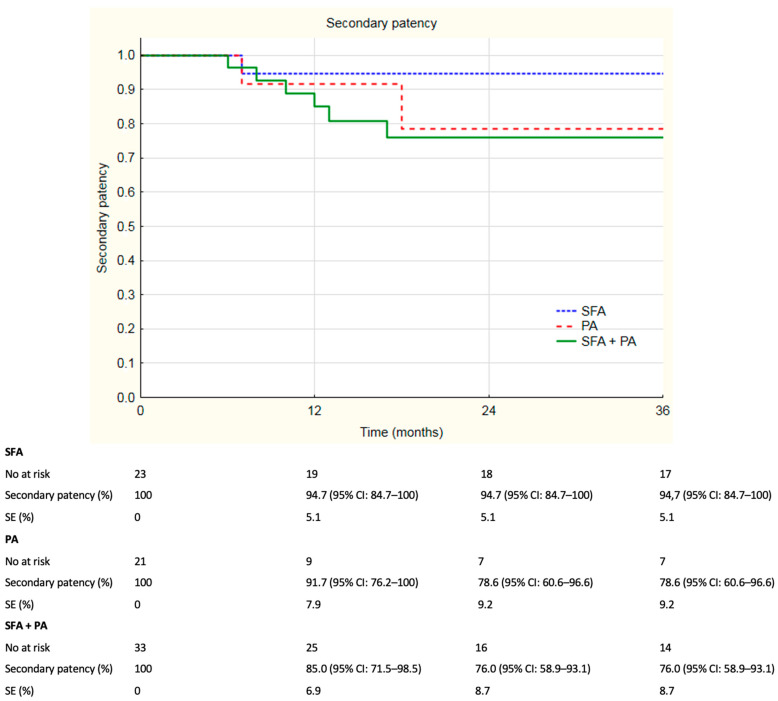
Kaplan–Meier estimates presenting secondary patency rates of Supera stent implantation by localization on duplex 36 months follow-up. SFA—superficial femoral artery, PA—popliteal artery, SE—standard error, CI—confidence interval.

**Table 1 jcm-12-04869-t001:** Baseline patients’ demographic and clinical characteristics.

Characteristics	Number of Patients n = 77	Percentage or Range
Male gender	58	75%
Age (years) (median)	71	11 (IQR)
BMI (kg/m^2^)	27.6 ± 4.9	21.2–39.4
HA	63	82%
Smoking	56	73%
Diabetes mellitus	42	54%
Hyperlipidemia	31	41%
Cerebrovascular disease	8	11%
CAD	26	34%
Renal insufficiency	17	22%
Rutherford classification		
2	2	3%
3	31	40%
4	10	13%
5	25	32%
6	9	12%
CLTI	44	57%

HA—arterial hypertension, CAD—coronary artery disease, CLTI—chronic limb-threatening ischemia, BMI—body mass index.

**Table 2 jcm-12-04869-t002:** Baseline lesions and stent characteristics.

Characteristics	Number	Percentage or Range
TASC classification		
A	3	4%
B	24	31%
C	24	31%
D	26	34%
ISR	19	25%
Lesion localization		
SFA	23	30%
PA	21	27%
SFA + PA	33	43%
Calcification		
None	4	5%
Mild	19	25%
Moderate	33	43%
Severe	21	27%
	Involved PA segment	
P1	12	16%
P2	28	36%
P3	14	18%
Number of patent outflow infra-popliteal vessels		
0–1	37	48%
2–3	40	52%
Number of stents per lesion		
1	62	81%
2	15	20%
3	0	0%
Lesion length (mm)	152.8 ± 94.6	20–400
CTO	51	66%
Stent diameter (median) (mm)	5.5	4–7
Stent diameter		
4 mm	12	13%
4.5 mm	20	22%
5 mm	17	18%
5.5 mm	30	33%
6.5 mm	9	10%
7 mm	4	4%
Stent length (median) (mm)	150	60–200

TASC—Trans-Atlantic Inter-Society Consensus Document, ISR—in-stent restenosis, SFA—superficial femoral artery, PA—popliteal artery, CTO—chronic total occlusion.

**Table 3 jcm-12-04869-t003:** Reinterventions in 36-month follow-up.

Reintervention	Number of Cases
PTA	4
PTA-DEB	2
Stent implantation	8
Thrombolysis	8
Femoropopliteal bypass	6
Atherectomy	3

PTA—percutaneous transluminal angioplasty, PTA-DEB—percutaneous transluminal angioplasty—drug eluting balloon.

## Data Availability

The data presented in this study are available on request from the corresponding author.

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
