# Peer review of "Three-Year Real-World Outcomes of Interwoven Nitinol Supera Stent Implantation in Long and Complex Femoropopliteal Lesions"

_jcm, 2023, doi:10.3390/jcm12144869_

Round 1

Reviewer 1 Report

Great experience with the Supera stent. Following are my suggestions to improve the manuscript.

Line 42: “…mortality and amputations…”: use MACE and MALE, here for the first time (instead that in line 193).

Lines 42,43: “…(CLTI) which is associated with increased 41 rates of mortality and amputations [2,3]. To prevent those aggravating complications proper treatment is required.”.

To better define the critical picture of CLTI, I would add: “To prevent those aggravating complications proper treatment is required. Nonetheless, the latter bears a considerable mortality in the short and medium term, given the high incidence of risk factors for atherosclerosis and coronary artery disease in patients with CLTI [Martelli E, et al. Sex-Related Differences and Factors Associated with Peri-Procedural and 1 Year Mortality in Chronic Limb-Threatening Ischemia Patients from the CLIMATE Italian Registry. J Pers Med. 2023 Feb 11;13(2):316. doi: 10.3390/jpm13020316].”

Line 73: you should at least make a comment on the indication to operate two patients with moderate intermittent claudication (Rutherford stage 2, table1).

Lines 73-74: …femoropopliteal…

Line 81: “…77. patients have been included to this retrospective, single center study”. This is a result of your study, and should be moved to the Results section of the manuscript.

Line 81, 174, 179: never start a sentence with a number: write the number in letters.

Line 17 vs 83: be consistent: decide if to use PAD, or LEAD.

Line 105: “…92 Supera stents were deployed”. This is a result of your study, and should be moved to the Results section of the manuscript. 

Line 167: correct the total of CLTI patients: 43 or 44? In table 1, move Rutherford classification to the left, and adjust the percentages of the classification (100%, not 101%).

Line 170: adjust the percentages of the stent diameters: the total must be 100%, not 101%.

Line 174: 143 patients…: looking at table 1, they are 144.

Lines 181-182: you should specify which zone of the PA artery you stented (P1, P2, or P3?), to better understand your results.

Lines 205-227: you should cite figures 2, 3, 4, and 5 after the patency results.

The legend of figure 2 is wrong: it is not reported that the Kaplan-Meier deals with survival, as reported in the graph.

Punctuation is poor throughout the manuscript: make it revise by a professional translator.

Punctuation is poor throughout the manuscript: make it revise by a professional translator.

Author Response

Thank you very much for the comprehensive review, it helped to improve our manuscript. Below we present our point-by-point response to your comments. 

Great experience with the Supera stent. Following are my suggestions to improve the manuscript.

Line 42: “…mortality and amputations…”: use MACE and MALE, here for the first time (instead that in line 193).

ANSWER: Changed

Lines 42,43: “…(CLTI) which is associated with increased 41 rates of mortality and amputations [2,3]. To prevent those aggravating complications proper treatment is required.”.

To better define the critical picture of CLTI, I would add: “To prevent those aggravating complications proper treatment is required. Nonetheless, the latter bears a considerable mortality in the short and medium term, given the high incidence of risk factors for atherosclerosis and coronary artery disease in patients with CLTI [Martelli E, et al. Sex-Related Differences and Factors Associated with Peri-Procedural and 1 Year Mortality in Chronic Limb-Threatening Ischemia Patients from the CLIMATE Italian Registry. J Pers Med. 2023 Feb 11;13(2):316. doi: 10.3390/jpm13020316].”

ANSWER: The sentence was added together with the reference.

Line 73: you should at least make a comment on the indication to operate two patients with moderate intermittent claudication (Rutherford stage 2, table1).

ANSWER: The comment was added, those were active and relatively young patients with life limiting intermittent claudication.

Lines 73-74: …femoropopliteal…

ANSWER: Changed

Line 81: “…77. patients have been included to this retrospective, single center study”. This is a result of your study, and should be moved to the Results section of the manuscript.

ANSWER: The placement of the sentence was changed.

Line 81, 174, 179: never start a sentence with a number: write the number in letters.

ANSWER: Changed.

Line 17 vs 83: be consistent: decide if to use PAD, or LEAD.

ANSWER: LEAD was changed into PAD in line 83.

Line 105: “…92 Supera stents were deployed”. This is a result of your study, and should be moved to the Results section of the manuscript.

ANSWER: This sentence was removed. Information about number of implanted stents is in results section.

Line 167: correct the total of CLTI patients: 43 or 44? In table 1, move Rutherford classification to the left, and adjust the percentages of the classification (100%, not 101%).

ANSWER: Corrected

Line 170: adjust the percentages of the stent diameters: the total must be 100%, not 101%.

ANSWER: Corrected

Line 174: 143 patients…: looking at table 1, they are 144.

ANSWER: Corrected

Lines 181-182: you should specify which zone of the PA artery you stented (P1, P2, or P3?), to better understand your results.

ANSWER: Additional data was provided regarding which segment of PA was stented in table 2.

Lines 205-227: you should cite figures 2, 3, 4, and 5 after the patency results.

ANSWER: We cited specific figures after patency results.

The legend of figure 2 is wrong: it is not reported that the Kaplan-Meier deals with survival, as reported in the graph.

ANSWER: Corrected from cumulative survival to ,,patency rates’’.

Punctuation is poor throughout the manuscript: make it revise by a professional translator.

ANSWER: We corrected the punctuation throughout the manuscript.

Reviewer 2 Report

This paper is a retrospective analysis of outcomes from implantation of Supera -stents. The procedures performed as well as results are well described. Nothing is mentioned about how patients were selected for the specific implantation of the stent. How accurate is the database in accurately finding all patients treated with Supera-stents in the femora-popliteal position. The authors need to provide information concerning the treatment strategy for lower limb ischemia, such as patient volume and treatments over time such as bypass-procedures. As it now stands it is difficult to understand why included patients had specific Supera-stents and not other possibilities such as DEB, debulking procedures etc.

Several information steps are missing such as;

outflow; patent infra-popliteal vessels, number etc?

Treatment strategy concerning anti platelet strategy or antikoagulation?

Reversal of clinical symptoms such as improved walking distance?

Limb salvage rate?

A large proportion of patients were smokers, any efforts to quit? Did any of the patients quit during the study period

The paper is very long and needs significant shortening.

Discussion is somewhat irrelevant and needs to focus on present results from the study and relate these findings to other studies and avoid phrases like ”especially good” etc

Figure 1 is unnecessary

avoid subjective phrases such as very good, especially good. Keep sentences short and avoid to much information in text if tables are present

Author Response

This paper is a retrospective analysis of outcomes from implantation of Supera -stents. The procedures performed as well as results are well described. Nothing is mentioned about how patients were selected for the specific implantation of the stent.

ANSWER: We provided more information about strategy of lower limb ischemia treatment and exact stent selection.

How accurate is the database in accurately finding all patients treated with Supera-stents in the femora-popliteal position.

ANSWER : All Supera stents implantation procedures were extracted from the database. Our Medical system enables to search all products that were used in the procedures performed in the department.

The authors need to provide information concerning the treatment strategy for lower limb ischemia, such as patient volume and treatments over time such as bypass-procedures. As it now stands it is difficult to understand why included patients had specific Supera-stents and not other possibilities such as DEB, debulking procedures etc.

ANSWER: We expanded information concerning the treatment strategy for lower limb ischemia in the methods section.

Several information steps are missing such as;

outflow; patent infra-popliteal vessels, number etc?

ANSWER: We provided data regarding number of the outflow vessels in table 2.

Treatment strategy concerning anti platelet strategy or antikoagulation?

ANSWER: Dual antiplatelet therapy was prescribed for at least 30 days in every patient. The information was added to the text.

Reversal of clinical symptoms such as improved walking distance?

ANSWER: As long we are able to provide data about limb salvage rate, we do not have enough data to support reversal of clinical symptoms. After additional analysis of the follow-up visits we established that information about reversal of the symptoms was not noted in a structured way. As a result, we cannot provide solid results and conclusions regarding information such as improved walking distance. However taking into consideration good patency rates and good limb salvage rate it might be supposed that reversal of clinical symptoms rate was sufficient. Nevertheless, it would be perfect to provide this data.

Limb salvage rate?

ANSWER: We performed additional analysis to provide limb salvage rate. At 36 months limb salvage rate was 84,7%.

A large proportion of patients were smokers, any efforts to quit? Did any of the patients quit during the study period

ANSWER: Every patient is being convinced to quit smoking during follow-up visits. However, we did not collect data on the number of patients that quitted smoking during the study period.

The paper is very long and needs significant shortening.

ANSWER: We removed Figure 1 and shortened the paper. However, the Journal of Clinical Medicine requires the text in original papers to be at least 4000 words.

Discussion is somewhat irrelevant and needs to focus on present results from the study and relate these findings to other studies and avoid phrases like ”especially good” etc

ANSWER: Phrases like ,,very good’’ and ,,especially good’’ were removed. Discussion was shortened and modified in order to focus on present results and relate the finding to different studies.

Figure 1 is unnecessary

ANSWER: Figure 1 was removed.

Comments on the Quality of English Language

avoid subjective phrases such as very good, especially good. Keep sentences short and avoid to much information in text if tables are present

ANSWER: Phrases like ,,very good’’ and ,,especially good’’ were removed. We made effort to shorten most long sentences and we provided more short sentences. We also removed some information that were repeated in text and tables.

Round 2

Reviewer 2 Report

This paper has now been extensively reviewed according to comments and much improved. I have no further comments

English is now ok

Author Response

Thank you for your comprehensive review. It enabled us to improve our paper.